# Architecture and Composition Dictate Viscoelastic Properties of Organ-Derived Extracellular Matrix Hydrogels

**DOI:** 10.3390/polym13183113

**Published:** 2021-09-15

**Authors:** Francisco Drusso Martinez-Garcia, Roderick Harold Jan de Hilster, Prashant Kumar Sharma, Theo Borghuis, Machteld Nelly Hylkema, Janette Kay Burgess, Martin Conrad Harmsen

**Affiliations:** 1Department of Pathology and Medical Biology, University Medical Center Groningen, University of Groningen, Hanzeplein 1 (EA11), 9713 GZ Groningen, The Netherlands; f.d.martinez.garcia@umcg.nl (F.D.M.-G.); r.h.j.de.hilster@umcg.nl (R.H.J.d.H.); t.borghuis@umcg.nl (T.B.); m.n.hylkema@umcg.nl (M.N.H.); j.k.burgess@umcg.nl (J.K.B.); 2W.J. Kolff Institute for Biomedical Engineering and Materials Science-FB41, University Medical Center Groningen, University of Groningen, A. Deusinglaan 1, 9713 AV Groningen, The Netherlands; p.k.sharma@umcg.nl; 3Groningen Research Institute for Asthma and COPD (GRIAC), University Medical Center Groningen, University of Groningen, Hanzeplein 1 (EA11), 9713 AV Groningen, The Netherlands; 4Department of Biomedical Engineering-FB40, University Medical Center Groningen, University of Groningen, A. Deusinglaan 1, 9713 AV Groningen, The Netherlands

**Keywords:** extracellular matrix, ECM hydrogel, viscoelasticity, decellularized organs, Maxwell model

## Abstract

The proteins and polysaccharides of the extracellular matrix (ECM) provide architectural support as well as biochemical and biophysical instruction to cells. Decellularized, ECM hydrogels replicate in vivo functions. The ECM’s elasticity and water retention renders it viscoelastic. In this study, we compared the viscoelastic properties of ECM hydrogels derived from the skin, lung and (cardiac) left ventricle and mathematically modelled these data with a generalized Maxwell model. ECM hydrogels from the skin, lung and cardiac left ventricle (LV) were subjected to a stress relaxation test under uniaxial low-load compression at a 20%/s strain rate and the viscoelasticity determined. Stress relaxation data were modelled according to Maxwell. Physical data were compared with protein and sulfated GAGs composition and ultrastructure SEM. We show that the skin-ECM relaxed faster and had a lower elastic modulus than the lung-ECM and the LV-ECM. The skin-ECM had two Maxwell elements, the lung-ECM and the LV-ECM had three. The skin-ECM had a higher number of sulfated GAGs, and a highly porous surface, while both the LV-ECM and the lung-ECM had homogenous surfaces with localized porous regions. Our results show that the elasticity of ECM hydrogels, but also their viscoelastic relaxation and gelling behavior, was organ dependent. Part of these physical features correlated with their biochemical composition and ultrastructure.

## 1. Introduction

The extracellular matrix (ECM) is the acellular component of all organs and tissues: a three-dimensional (3D) mixture of proteins embedded in a gel of water-retaining negatively charged polysaccharides such as glycosaminoglycans (GAGs) [1]. While the ECM composition is tissue-specific, its components and organization can vary among structures within the same organ [2,3]. The ECM guides cell fate and provides mechanical support to the cells embedded within [4,5].

Historically, ECM mechanics were solely evaluated in terms of elasticity (i.e., elastic modulus often denoted by *E*, and also called stiffness)—the resistance of an object to undergo reversible deformation (strain, *ε*) in response to applied force (Stress, *σ*) [6,7]. In purely elastic materials, the mechanical energy is stored as strain and the elastic modulus remains strain rate independent [8]. Strain rate ($\overset{\cdot }{\varepsilon }$) is the speed with which a material is compressed. Due to variations in ECM composition and organization, the response to mechanical stress and strain differs among organs [9]. Nevertheless, more recent studies have shown that due to a large water content, the ECM is not elastic but viscoelastic in nature [7,10,11], where viscosity plays an active role in matrix mechanics [7]. Viscosity is a material property that arises from the resistance of a fluid to deformation. The combination of both elastic and viscous responses leads to a time-dependent stress dissipation (i.e., stress relaxation), a phenomenon known as viscoelasticity [8]. Unlike purely elastic materials that store and retain energy, a viscoelastic material will dissipate energy in the presence of stress over time, making the elastic modulus strain rate dependent. Thus, viscoelasticity is an inherent property of the ECM, that has only recently been recognized within biological systems [7,10,11].

The common in vitro systems to mimic in vivo ECM are hydrogels: water-swollen polymeric networks [12,13,14,15]. The viscoelasticity of hydrogels is tailored by varying the type and molecular weight of the constituent polymers, their concentration, and the crosslinking conditions (e.g., temperature, pH) [16,17,18,19,20]. Using hydrogels as ECM mimics illustrates that the cells perceive the surrounding viscoelasticity by applying a pushing or pulling force and sensing the time-dependent deformation response from the environment [10,21].

Hydrogels can be prepared from individual ECM components such as collagen, elastin or non-sulphated GAGs such as hyaluronic acid [18,19,22,23]. These homopolymeric hydrogels demonstrate the influence of individual matrix components in fundamental aspects of cell biology [18,19]. While collagen and elastin are the major load-bearing and elastic ECM components, other molecules, such as GAGs, hold on to water and play a role in matrix mechanics by offering resistance to the mobility of water within the ECM [24]. The main role of GAGs in ECM mechanics is that of lubrication and stress absorption. While homopolymeric networks demonstrate the contribution of individual components in matrix-cell biology, a factual representation of matrix viscoelasticity requires the presence of the native, heterogeneous components from the ECM. Thus, using organ-derived ECM might provide a biomimetic model that, due to their source, retains native ECM components involved in matrix mechanics [25].

We set out to compare three organs that are continuously subjected to mechanical forces but differ in function, i.e., the skin, lung and left ventricle (LV) of the heart. Skin is pliable and deformed due to body movement, while the lungs undergo inflation/deflation cycles via the action of the diaphragm and compression of the chest. Finally, the heart is a continuously beating muscle. The molecular composition of the ECM shares the presence of collagen type I, while organ-specific differences exist that relate to the function of the skin, lung and LV. We hypothesized that the composition and architecture of the ECM of skin, lung and LV dictates their mechanical properties in particular viscoelasticity. We set out to test this hypothesis using hydrogels from decellularized skin, lung and LV.

## 2. Materials and Methods

### 2.1. Decellularization

Porcine skin, lungs and hearts (~6-month, female) were purchased from a local slaughterhouse (Kroon Vlees, Groningen, the Netherlands). The heart and skin were decellularized as described previously [26]. The lung was decellularized as described by Pouliot et al. [27] with the exception that in this case the lungs were finely blended prior to decellularization.

The dissected LVs from pig heart and skin were dissected into 1 cm^3^ cubes. The tissues were washed with 1x Dulbecco’s phosphate-buffered saline (DPBS; BioWhittaker^®^, Walkersville, MD, USA) at room temperature (RT) and then minced in a commercial Bourgini 21.3001 blender (Bourgini, Breda, the Netherlands) with DPBS until the pieces were ~1 mm^3^ in size. After a second DPBS wash, the tissue homogenate was sonicated for 1 min at 100% power. After sonication, the tissue homogenate was washed with DPBS again and incubated in 0.05% trypsin (Thermo Fisher Scientific, Waltham, MA, USA) at 37 °C for 3 h under constant shaking. After trypsin treatment, tissue material was washed again with DPBS and frozen at −20 °C for at least 24 h.

The homogenate was thawed, washed with Milli-Q^®^ water for 3 h and then sequentially treated with saturated NaCl (6 M) solution, 70% ethanol, 1% SDS solution (Sigma-Aldrich, St. Louis, MO, USA), 1% Triton X-100 (Sigma-Aldrich), 1% sodium deoxycholate (Sigma-Aldrich) and 30 µg/mL DNase (Sigma-Aldrich) (in MgSO_4_ 1.3 mM and CaCl_2_ 2 mM), with three washes with Milli-Q^®^ water between treatments, 24 h each at RT with constant shaking, except for the enzymatic treatments, which were at 37 °C while shaking. The resultant decellularized ECM was stored in sterile DPBS containing 1% penicillin-streptomycin (Gibco Invitrogen, Carlsbad, CA, USA) at 4 °C.

The lung was dissected, and the cartilaginous airways removed before cutting into ~1 cm^3^ cubes and minced until it was ~1 mm^3^ in size with a commercial blender. The resulting tissue homogenate was then repeatedly washed with Milli-Q^®^ water and spun down at 3000× *g* until the supernatant cleared completely. The remaining tissue homogenate went through two rounds of sequential treatment with 0.1% Triton X-100, 2% sodium deoxycholate, 1 M NaCl solution and 30 µg/mL DNase in 1.3 mM MgSO_4_ and 2 mM CaCl_2_, 10 mM Tris pH 8 (Sigma-Aldrich) solution each for 24 h at 4 °C with constant shaking, except for the enzymatic treatments, which were at 37 °C with shaking. Between treatments, the homogenate was washed three times with Milli-Q^®^ water, centrifuged at 3000× *g* between washes. After two cycles of decellularization, the tissue homogenate was sterilized by adding 0.18% peracetic acid and 4.8% ETOH, and left shaking at 4 °C for 24 h. After tissue sterilization, the resultant decellularized ECM was stored in sterile DPBS containing 1% penicillin-streptomycin at 4 °C (Figure 1a).

### 2.2. Hydrogel Preparation

The blended, decellularized skin-ECM, lung-ECM and LV-ECM were snap-frozen in liquid nitrogen and lyophilized with a FreeZone Plus lyophilizer (Labconco, Kansas City, MO, USA) and then ground into a powder with an A11 Analytical mill (IKA, Staufen, Germany). Then, 20 mg/mL of ECM powder was digested with 2 mg/mL porcine pepsin (Sigma-Aldrich) in 0.01 M HCl under constant agitation at RT either for 8 h (LV-ECM) or 48 h (lung-ECM and skin-ECM) (Figure 1b). After digestion, the pH was neutralized with 0.1 M NaOH and brought to 1× DPBS with one-tenth volume 10× DPBS to generate the so-called ECM pre-gel solution. For hydrogel formation, pre-gel from each organ-derived ECM was poured in a mold and incubated at 37 °C for 1 h (Figure 1a). After gelation, the hydrogels were equilibrated in HBSS medium (Lonza, Bazel, Switzerland) and incubated for 24 h prior to experiments.

### 2.3. Turbidity Assay

The gelling kinetics of skin-ECM, lung-ECM and LV-ECM hydrogels were analyzed with a turbidimetric assay [28,29,30]. The ECM pre-gel solutions were pipetted (150 µL) into a precooled (4 °C) 96-well plate (Corning Inc., Corning, NY, USA). The cooled 96-well plate containing the pre-gels was loaded into a pre-heated (37 °C) CLARIOstar Plus multi-mode microplate Reader (BMG Labtech, Ortenberg, Germany), and the absorbance measured at 405 nm with 30-second intervals for 2 h. Absorbance values were normalized with the following formula:(1)$$NA=\frac{\left(A-A_{min}\right)}{\left(A_{max}-A_{min}\right)}\times 100\%$$
where *NA* is the normalized absorbance, *A* is the absorbance at any given time, *A_min_* is the lowest observed absorbance and *A_max_* is the maximal absorbance. The normalized curves were plotted to start from gelation, omitting the lag time. From the sigmoidal-shaped turbidity curves, we calculated the following kinetic parameters: *A_min_* and *A_max_*, *T_lag_* (the time value at which the normalized absorbance is 0), *T*_1/2_ (the time at which the normalized absorbance is 50%), *T_end_* (the time at which the normalized absorbance is 100%) and *S* (the slope of the linear portion of the curve), indicating the speed of gelation. Three independent turbidity measurements were performed with three replicates each (*n* = 3). 

### 2.4. Viscoelastic Properties

The elastic modulus and stress relaxation properties of skin-ECM, lung-ECM and LV-ECM hydrogels were evaluated on the Low-Load Compression Tester (LLCT) as described previously [26,31,32]. Data were acquired with LabVIEW 7.1, and subsequently analyzed with MatLab 2018 (MathWorks^®^ Inc., Natick, USA). Hydrogels (300 µL) underwent uniaxial compression with a 2.5-millimeter diameter plunger at three different locations, leaving 2 mm from the edge and 2.5 mm between each compression site. When compressed, each hydrogel reached 80% of its original thickness (i.e., strain, *ε* = 0.2) at a strain rate ($\overset{\cdot }{\varepsilon }$) of 0.2 s^−1^ (or a deformation rate of 20%·s^−1^) at room temperature (≅25 °C). The elastic modulus was determined as the slope between the stress–strain curve.

After compression, the strain was kept constant (0.2) for 100 s, and the stress was continuously monitored. The time to reach 50% stress relaxation was determined by comparing the stress relaxation percentage at *t* = 0 s and *t* = 100 s. The relaxing stress as a function of time (*σ*(*t*)) was divided by the constant strain of 0.2 to obtain the value of continuously decreasing modulus *E*(*t*). Data were acquired according to a generalized Maxwell model, Equation (2), to calculate the values of *E_i_* and *τ_i_* for individual Maxwell elements, where *i* varies from 1 to *n*. The *τ_i_* is the relaxation time constant for each individual Maxwell element and is the ratio of *η**_i_* (dashpot) and *E_i_* (spring) for that element (Figure 1). The number of Maxwell elements necessary to fit the experimental data were determined by visually fitting a plot that shows the decrease in Χ^2^ value with the addition of every extra Maxwell element (Figure 1b). The required number of Maxwell elements were chosen when no further decrease in Chi^2^ was observed (Figure 1b).
(2)$$E\left(t\right)=E_{1}e^{-t/\tau_{1}}+E_{2}e^{-t/\tau_{2}}+E_{3}e^{-t/\tau_{3}}+\ldots \;E_{n}e^{-t/\tau_{n}}$$

The relative importance (*R_i_*) of each Maxwell element in terms of percentage within the relaxation process was expressed as the proportion of its spring constant, *E_i_*, to the sum of all spring constants Equation (3).
(3)$$R_{i}=100\times \frac{E_{i}}{\sum_{i=1}^{n}E_{i}}$$

### 2.5. Protein Quantification

Total protein content was quantified with a Pierce™ Modified Lowry Protein Assay Kit (Thermo Fisher Scientific). For this, 2 µL of pre-gel solution was diluted in 38 µL of 1x DPBS and transferred to a well in a non-adhesive Costar^®^ 96-well plate (Corning Inc., Kennebunk, ME, USA). Next, 200 µL of modified Lowry protein assay was added per well before incubation at RT for 10 min. Fresh 1 N Folin-Ciocalteu’s phenol reagent was prepared by diluting 2 N Folin-Ciocalteu’s phenol reagent with an equal volume of Milli-Q^®^ water and 20 µL of this solution was added per well and incubated at RT for 30 min. The absorbance was read at 750 nm with Benchmark Plus™ microplate spectrophotometer system (Bio-Rad, Hercules, CA, USA). The protein concentration was determined based on a calibration curve derived from a dilution series of bovine serum albumin (Thermo Fisher Scientific). DPBS served as absorbance blanks. The protein concentration (µg/mL) from each organ-derived ECM was calculated from four independent experiments each performed in triplicate (Figure 1d).

### 2.6. Sulphated Glycosaminoglycans (sGAGs) Quantification

Total sGAGs content was quantified with a 1,9-Dimethyl-Methylene Blue zinc chloride double salt (DMMB) assay based on reported protocols [26,33,34]. For this, 20–25 mg of ECM powder was incubated in 300 µL of 75 mM NaCl, 25 mM EDTA, 50 µL of 10% SDS, and 5 µL of proteinase K (19.9 mg/mL) (Thermo Scientific) at 60 °C overnight. Next, 20 µL of digested organ-derived ECM was mixed with 200 µL of DMMB solution (comprising 19 mg DMMB in 40 mM Glycine, 38 mM NaCl, 100 mM acetic acid pH 3) and the absorbance read at 525 nm immediately. Serial dilutions of chondroitin sulphate (Sigma-Aldrich) were used for the calibration curve, and the absorbance blank corrected with DMMB solution. The total sGAGs content (µg/mL) from each organ-derived ECM was calculated from four independent experiments each performed in triplicate (Figure 1d).

### 2.7. Histological Characterisation

Hydrogels were fixed with 2% formalin for 24 h at 4 °C. All samples were conventionally processed for histology using a graded alcohol to dehydrate and paraffin embedded. Sections of 4 µm thickness were deparaffinized and stained with Alcian Blue (pH 2.5) to visualize GAGs [35] and Masson’s Trichrome (MTC) and Picrosirius Red (PSR) to visualize collagens [36,37], following the previously cited protocols. Slides were covered with Permount™ Mounting Medium (Fisher Chemical™, Waltham, MA, USA) (Figure 1e). Analyses derived from histology staining are detailed in Section 2.9.

### 2.8. Immunohistochemistry

Sections (4 µm) were deparaffinized and incubated with citrate buffer for antigen retrieval. After blocking to prevent non-specific background staining, sections were incubated with 1 µg/mL of mouse anti-human COL1A1 (Abcam, Cambridge, UK) or 1 µg/mL of goat anti-human elastin (Cedarlane, Burlington, VT, USA), respectively, at 4 °C, overnight. After 3 DPBS washes, sections were incubated with a 1:100 dilution of an anti-mouse horse radish peroxidase conjugate (Dako, Santa Clara, CA, USA) or a 1:100 dilution of an anti-goat horse radish peroxidase conjugate (Dako) at RT for 1 h. The staining was then visualized with Vector^®^ NovaRED™ (Vector Laboratories, Burlingame, USA). Slides were counterstained with Hematoxylin and covered with Permount^®^ mounting media (Figure 1e).

### 2.9. Imaging and Image Analysis

All stained sections were scanned with a Slide Scanner (Hamamatsu Photonics K.K., Herrsching, Germany) at 20× magnification (Figure 1e). PSR fluorescent images (PSR-fluo) were generated with Zeiss LSM 780 CLSM confocal microscope (Carl Zeiss NTS GmbH, Oberkochen, Germany), λ_ex_ 561 nm/λ_em_ 566/670 nm at 40× magnification [38]. COL1A1, Elastin scans and PSR-fluo images were analyzed with TWOMBLI, an ImageJ/Fiji [39] plugin to quantify patterns in ECM [40]. Before analyzing the COL1A1 and Elastin, the Vector^®^ NovaRED™ color was isolated from the images using a color deconvolution plugin [41]. The images with only Vector^®^ NovaRED™ color were subsequently used for the analysis.

TWOMBLI was used to determine the number of fibers, end points, branching points, total fiber length and alignment, lacunarity (number and size of gaps in the matrix) and high-density matrix proportion (measure for compactness of matrix).

### 2.10. Hydrogel Ultrastructure

Hydrogel ultrastructure was visualized with scanning electron microscopy (SEM). First, all hydrogels were fixed with a 1% paraformaldehyde, 1% formalin at 4 °C for 24 h. Then, the hydrogels were washed three times with DPBS and once with Milli-Q^®^ water to remove any remaining fixatives and salts. The hydrogels were plunged in liquid nitrogen and freeze-dried. Dried samples were glued on top of 0.5” SEM pin stubs (Agar Scientific, Stansted, UK) and Au-Pd coated after rinsing with Argon with Leica EM SCD050 sputter coater device (Leica Microsystems B.V., Amsterdam, Netherlands). Hydrogels were visualized at 5000× and 10,000× magnification, at 3.0 kV with Zeiss Supra 55 STEM (Carl Zeiss NTS GmbH) (Figure 1c).

### 2.11. Statistical Analysis

All statistical analyzes were performed using GraphPad Prism v9.1.0 (GraphPad Company, San Diego, USA). All data were scrutinized for outliers using the robust regression and outlier removal (ROUT) test and analyzed for normality using Shapiro–Wilk and D’Agostino and Pearson tests [42,43,44]. Based on this, LLCT data were analyzed with Kruskal–Wallis and Dunn’s post hoc test and with one-way ANOVA and Tukey’s post hoc test. Lowry, TWOMBLI and turbidity data were also analyzed with one-way ANOVA and Tukey. DMMB data were analyzed with Student’s *t*-test. Graphs are presented as median with quartiles or mean values with standard deviation (SD). All *p* values below * 0.05; ** 0.01; *** 0.001 and **** 0.0001 were considered statistically significant.

## 3. Results

### 3.1. Turbidity

Measuring the gelation kinetics of ECM hydrogels using turbidimetric analysis is based on the increased turbidity during gelling, i.e., increased absorbance. The quantitative breakdown of the following parameters from these curves: *A_min_*, *A_max_*, *T_lag_*, *T*_1/2_, *T_end_* and *S*, is shown in Table 1.

All the organ-derived ECMs gelated in a sigmoidal pattern that started after a lag period of 10–13 min (Figure 2). The minimum and maximum absorbance (*A_min_* and *A_max_*) remained highest in the skin-ECM, while the lung-ECM and the LV-ECM were lower and comparable. The skin-ECM sol-gel transition was faster than the lung-ECM and the LV-ECM. The total gelling time (*T_end_*) was, therefore, the shortest in the skin-ECM, followed by the LV-ECM and finally the lung-ECM (Figure 2, Table 1).

### 3.2. Elastic Modulus and Stress Relaxation

The elastic modulus of the skin-ECM (1.66 ± 0.82 kPa) was lower than the lung-ECM (4.98 ± 1.81 kPa) and the LV-ECM (4.38 ± 1.73 kPa) (Figure 3a).

The time to reach 50% stress relaxation was fastest in the skin-ECM (5.16 ± 4.57 s), compared to the lung-ECM (49.40 ± 4.35 s) and lastly the LV-ECM (51.63 ± 1.18 s). The elastic modulus or stress relaxation of the LV-ECM and the lung-ECM did not differ (Figure 3b,c).

### 3.3. Maxwell Analysis

Maxwell analysis showed differences in the relative importance (*R_i_*) and the time each Maxwell element remains active (tau; *τ*) among organ-derived ECM hydrogels.

The fastest or first (1st) Maxwell element had a greater *R_i_* in the skin-ECM (66.42 ± 9.07%), than both the lung-ECM (53.02 ± 3.39%) and the LV-ECM (53.48 ± 4.72%). The intermediate or second (2nd) Maxwell element, had also a greater *R_i_* in the skin-ECM (33.58 ± 9.07%) than in the LV-ECM (33.58 ± 9.07%). The slow or third (3rd) Maxwell element was not detected in the skin-ECM but had a lower *R_i_* in the lung-ECM (16.29 ± 2.15 s) than in the LV-ECM (19.05 ± 3.29 s) (Figure 4a).

The *τ*_1_ from first 1st Maxwell element remained active for less time in the skin-ECM (0.26 ± 0.08 s) than in the lung-ECM (0.33 ± 0.07 s) and the LV-ECM (0.34 ± 0.06 s). The *τ*_2_ was also active for a shorter time in the skin-ECM (2.63 ± 2.25 s) than in the lung-ECM (3.88 ± 1.60 s) and the LV-ECM (4.78 ± 1.35 s). No differences were found between the *τ*_3_ of the lung-ECM (40.41 ± 21.33 s) and the LV-ECM (51.24 ± 18.14 s) (Figure 4b).

### 3.4. Protein and sGAGs Content

The protein content of the pre-gels of the skin-ECM (973 ± 207 µg/mL), lung-ECM (1029 ± 154 µg/mL) and LV-ECM (912 ± 98 µg/mL) did not differ (Figure 5a). The pre-gels of the skin-ECM (202 ± 39 µg/mL) had significantly higher sGAGs contents than the LV-ECM (11 ± 10 µg/mL ****). In the lung-ECM, the sGAGs were below the detection limit of the DMMB assay Figure 5b).

### 3.5. Histologic Assessment

Histological staining of the skin-ECM, the lung-ECM and the LV-ECM with Alcian Blue Masson’s Trichrome (MTC) and Picrosirius Red (PSR) is shown in Figure 6.

All the hydrogels contained detectable levels of proteins and sGAGs that were arranged as fibrous meshworks. The skin-ECM had markedly higher levels of sGAGs by Alcian Blue compared to the lung-EMC and the LV-ECM, which had comparable levels.

The collagen networks in the EMC hydrogels, as visualized with MTC and PSR, showed differences. Collagen was observed as dense, condensed fibrous networks in the LV-ECM (MTC and PSR) and the skin-ECM (PSR). The lung-ECM showed a more finely distributed network of collagen fibers that was intermediate of the skin -ECM and the LV-ECM, which had also bound less dye both in the MTC and PSR stains. The LV-ECM was heterogeneous with large interfibrous areas of irregular size, while both the lung-ECM and the skin-ECM appeared more homogeneously organized. Interestingly, this architecture showed more prominently in the skin-ECM after PSR staining because MTC staining showed a more irregular binding.

### 3.6. Matrix Organisation

Both MTC and PSR predominantly stain collagen-type fibers. This was corroborated by the immunohistochemistry for COL1A1, a component of the major tissue collagen, i.e., type I (Figure 7). The collagen I architecture of the LV-ECM showed condensed fibers surrounding large irregularly shaped voids. In contrast, in the lung-ECM and the skin-ECM, the collagen I architecture was comprised of a fine reticular meshwork. The skin-ECM had a higher collagen I content than the lung-ECM. The elastin was distributed in condensed patches in all the ECM hydrogels (Figure 7). It would appear that the LV-ECM contained higher levels of elastin than the skin-ECM and the lung-ECM.

Fluorescent imaging of PSR-stained sections (Figure 8) was used to run detailed analyses of the collagen fibers with respect to size, shape and organization (Twombli, Table 2).

#### 3.6.1. Number of Fibers and Length (Mean and Total)

The mean and total number and length of the immuno-stained collagen type I fibers differed between all three organ-derived ECM hydrogels (Figure 7, Table 2). The skin-ECM had less elastin fibers, that were also shorter compared to the LV-ECM. The lung-ECM had shorter elastin fibers than the LV-ECM (Figure 7, Table 2). Histochemical picrosirius fluoro-micrographs revealed less fibers in the skin-ECM and the LV-ECM compared to the lung-ECM (Figure 8, Table 2). The mean fiber length of the lung-ECM was longer compared to the skin-ECM and the LV-ECM (Figure 7, Table 2).

#### 3.6.2. Branch Points and End Points

The number of immuno-stained collagen type I branch points and end points did not differ between the organ ECM hydrogels (Figure 7, Table 2). The number of immuno-stained elastin branch points and end points were both lower in the skin-ECM and the lung-ECM, compared to the LV-ECM (Figure 7, Table 2). Histochemical picrosirius fluoro-micrographs also showed similar numbers of fiber branch points in all three ECM hydrogels (Figure 8, Table 2). Yet, the number of end points was lower in the skin-ECM than in the lung-ECM and the LV-ECM. Additionally, the lung-ECM had less end points than the LV-ECM (Figure 8, Table 2).

#### 3.6.3. Lacunarity and High-Density Matrix (HDM)

The lacunarity of the collagen A1 distribution (Figure 7) did not differ between the ECM hydrogels from the skin, lung and LV, while the high-density matrix was larger in the skin-ECM than both the lung-ECM and the LV-ECM. The lacunarity of the elastin distribution (Figure 7) was higher in the skin-ECM and the lung-ECM compared to the LV-ECM. In contrast, the high-density matrix was smaller in both the skin-ECM and the lung-ECM compared to the LV-ECM. In the Picrosirius red-stained fluoro-micrographs (Figure 8), the lacunarity was smaller in the skin-ECM and the LV-ECM than in the lung-ECM. In the Picrosirius red-stained micrographs, the high-density matrix was larger in the skin-ECM than in the lung-ECM and the LV-ECM.

### 3.7. Ultrastrucure

The ultrastructure of hydrogels, visualized with SEM, showed qualitative differences between the skin-ECM and both the lung-ECM and the LV-ECM. In both the lung-ECM and the LV-ECM, most of the surface displayed a sheet-like organization, with randomly scattered openings. Beneath these sheets, a fibrous network could be discerned. In contrast, the skin-ECM lacked these condensed sheets and was comprised of a fibrous network with irregularly shaped pores with fibrils of a similar thickness (Figure 9).

## 4. Discussion

Our study shows that the ECM hydrogels from the skin, lung and LV of the heart, at an equal protein concentration, differ distinctly in composition, gelling kinetics and viscoelasticity. The mechanical properties and ultrastructure of the lung-ECM and the LV-ECM were similar. The content of sGAG was higher in the skin-ECM than the LV-ECM, while sGAGs were below the detection limit in the lung-ECM. Turbidity assays demonstrated that the skin-ECM had a higher starting absorbance and a faster sol-gel transition than in the lung-ECM and the LV-ECM. The skin-ECM hydrogels had a lower elastic modulus and faster stress relaxation than the lung-ECM and the LV-ECM. The surface topography of the skin-ECM hydrogels was more porous than the lung-ECM and the LV-ECM and the TWOMBLI analyses illustrated differences in the collagen and elastin fiber-related parameters (length, density, number, among others).

The first indication of an organ-specific ECM composition and ultrastructure is the difference in the time needed for pepsin digestion (skin, 8 h; lung and LV, 48 h). Depending on the composition, crosslinking and, consequently, the ultrastructure of the ECM, it will require more or less “cutting” by pepsin in order to be solubilized. This difference in pepsin digestion times as well as the decellularization methods required has been reported to be organ specific [45]. The decellularization process as well as the pepsin solubilization have been shown to impact the composition and the mechanical properties of ECM hydrogels and need to be optimized to each tissue or organ [38,39].

All the organ-derived ECM hydrogels had a unique turbidity profile and kinetic parameters, which implies that the composition of these ECM hydrogels differ with respect to their assembly mechanism and kinetics. The differences in the composition of the ECM can impact the gelling process, where the addition of the proteoglycan decorin increased the lag phase of collagen 1 gelation and collagen V can regulate the fiber diameter of collagen [46,47]. The increased total gelling time in the lung and LV-ECM could be due to the loss of collagen 1 telopeptides with longer pepsin digestion times, which is observed [30,48]. Turbidity analyses provide an insight into gelling kinetics but do not give information on the molecular assembly and fibril ultrastructure, the endpoint of which we evaluated using SEM and staining.

The elastic modulus of the skin-ECM hydrogels was 50% lower than the lung-ECM and the LV-ECM, consistent with previously reported data [26,31]. The stress relaxation of the skin-ECM was an order of magnitude faster than the lung-ECM and the LV-ECM. Stress dissipation occurs via the displacement of water and also via polymer network rearrangements. The speed and kinetics of each process can be mathematically modelled using Maxwell analysis [49]. Each rearrangement process is represented by a Maxwell element active for a definite time period (tau; *τ*) and has a definite contribution to the overall relaxation process, i.e., the relative importance (*R_i_*). A previous study showed that the stiffness and viscoelastic properties of ECM hydrogels still resemble that of the organ of origin [31]. The incorporation of cells into hydrogels may help bridge the gap, the small difference between tissue and ECM hydrogel mechanics, as well as to explain their contribution to the viscoelastic relaxation process of organs and tissues. Interestingly, the ECM hydrogels from the two organs that experience continuous rhythmic mechanical stresses had similar stiffnesses and viscoelasticity, in contrast to skin.

Maxwell analyses showed that the skin-ECM had only two Maxwell elements, while the lung-ECM and the LV-ECM had three. The skin-ECM’s first and second Maxwell elements (fast and intermediate) had a greater *R_i_* with a lower *τ* than in the lung-ECM and the LV-ECM. These elements are associated with the fastest stress dissipating components of the ECM, such as water and small molecules (e.g., growth factors). GAGs are among the ECM molecules that strongly bind, and thus resist the displacement of, water [50]. The quantification of sGAGs indicated a greater presence in the skin-ECM than in the LV-ECM but was below detection level in the lung-ECM.

Information about the role of (s)GAGs in organ and tissue viscoelasticity is limited. One study showed that stress relaxation decreases in GAG-depleted arteries [24]. Depleting sGAGs such as chondroitin sulphate, dermatan sulfate and heparan sulfate in lung tissue was shown to increase stress relaxation, whereas depleting hyaluronic acid had no effect [51]. No sGAGs were detected in the lung-ECM according to the DMMB assay, but Alcian Blue staining did detect the presence of GAGs in general. This finding might indicate that the non-sulfated GAG hyaluronic acid (HA) is an abundant GAG in lung-ECM hydrogels. In native lung tissue, HA is the most abundant GAG [52]. HA would not be detected using the DMMB assay owing to its non-sulfated nature [33]. Other authors have reported that detergent-based lung decellularization causes significant loss of both types of GAGs, as well as the depletion of specific sulfation patterns [53]. Our findings also indicate a loss in lung-ECM-derived sGAGs. Further studies should expand on the proteomic characterization of an organ-derived ECM, both from the original organ, as well as the resulting ECM after decellularization through mass-spectrometry. This information might give insights to the contribution of individual matrix components to viscoelasticity.

The differences among the hydrogels may not only depend on the composition but also on the architecture and conformation of the matrix. TWOMBLI analyses of the immunohistochemistry showed that COL1A1 had a higher density in the skin-ECM hydrogels than the lung-ECM and the LV-ECM. In Elastin, the TWOMBLI analyses showed differences in the fiber number, length, amount and density, among others, in the LV-ECM, compared to the skin-ECM and the lung-ECM. Observing the polymer network under SEM demonstrated that the surface microarchitecture of the LV-ECM and the lung-ECM hydrogels had a condensed layer that displayed a sheet-like organization. Localized regions showed the presence of pores on the surface. In contrast, the skin-ECM hydrogels had an open fiber structure with large fibers and pores. These findings correlate to the mechanical properties observed, as the lung-ECM and the LV-ECM had similar viscoelastic and surface architecture properties, while the skin-ECM was highly porous with lower stiffness and faster stress relaxation. Porosity represents a percentage of void space in a solid [54,55], where an excess of voids can compromise the mechanical stability of materials [56]. The swelling of hydrogels might have resulted in a different water content depending on the organ source of ECM. We recognize that not evaluating this is a limitation of our study. GAGs and proteoglycans contribute to the water retention of the ECM. The compositional differences in these proteins may affect the swelling and, consequently, also the mechanical properties of the ECM hydrogels.

Overall, these findings demonstrate that organ-specific ECM composition idiosyncrasies remain present after decellularization. Discrepancies exist with our colleagues, where a fourth Maxwell element in LV-ECM hydrogels is reported [26]. Compared to the protocol from our colleagues, we employed two hours more for pepsin digestion [26]. The pepsin digestion time influences organ-derived ECM hydrogel mechanics, which could explain the differences observed [29].

The analyses of hydrogel architecture through SEM can produce artifacts because of the required sample drying during the preparation process that sample fixation before freeze-drying may not prevent [57,58,59]. To corroborate the SEM results, we used fluorescent imaging of the picrosirius red stained sections. A possible way to image the ultrastructure of ECM hydrogels in a wet state would be 5(6)-Carboxytetramethylrhodamine N-succinimidylester (TAMRA-SE) staining in combination with confocal laser scanning microscopy [60,61]. Limitations concerning the LLCT have been addressed in the past by our research group [31].

## 5. Conclusions

In conclusion, organ-derived ECM hydrogels retain their specific composition and, with that, the accompanying mechanical properties and ultrastructure. Organ- or tissue-specific ECM hydrogels provide opportunities for simulating the organ or tissue microenvironment, opening possibilities for use in tissue engineering and as model systems for understanding disease underlying mechanisms. Organ ECM hydrogels enable the generation of novel models for mimicking and incorporating a native organ ECM in a research environment. Further characterizing the ECM composition of organs and ECM hydrogels will allow us to discern how different ECM proteins influence the mechanics, as well as what compositional elements ECM hydrogels need to more completely mimic the native environment. With organ-specific ECM hydrogels, we can explore cell–matrix interactions, which are dynamic and reciprocal. Both factors influence the biomechanical outcome of the cellular environment, leading to changes in both cell fate and ECM composition and mechanics. The crosstalk between cells and the ECM is often dysregulated in disease and, with organ-specific ECM hydrogels, we might elucidate the precise role of the ECM in the pathophysiology.

## Figures and Tables

**Figure 1 polymers-13-03113-f001:**
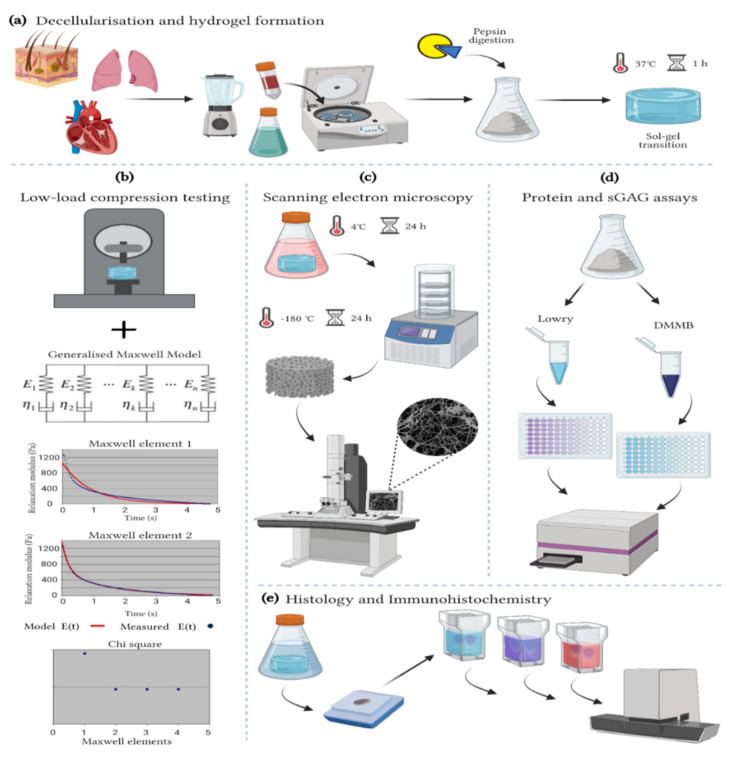
Methods. (**a**) Decellularization and hydrogel formation of skin-ECM, lung-ECM and LV-ECM (**b**) Low-load compression testing at 20% strain for 100 s. The data analyzed with a generalized Maxwell model of viscoelasticity. The number of Maxwell elements were determined based on curve-fitting the stress relaxation data (Relaxation modulus; Pa). The figure shows skin-ECM data, where two Maxwell elements were sufficient to explain their viscoelasticity, confirmed by the decrease in Chi^2^. (**c**) Scanning electron microscopy (SEM). All hydrogels were fixed for 24 h in 2% glutaraldehyde and 2% paraformaldehyde (1:1 ratio) at 4 °C, freeze-dried for 24 h, metal coated and visualized with SEM. (**d**) Protein and sulphated glycosaminoglycans (sGAGs) quantification with Lowry and DMMB assays. (**e**) Histology and Immunohistochemistry. Hydrogels were fixed for 24 h in 2% formalin, processed conventionally with a graded ethanol series for dehydration, paraffin embedded and sectioned. Sections (5 µm) were stained with Alcian Blue, Picrosirius Red (PSR) and Masson’s Trichrome (MTC) as well as immune stained for collagen type I (COL1A1) and Elastin and scanned with a Hamamatsu section scanner. (Figure created with BioRender).

**Figure 2 polymers-13-03113-f002:**
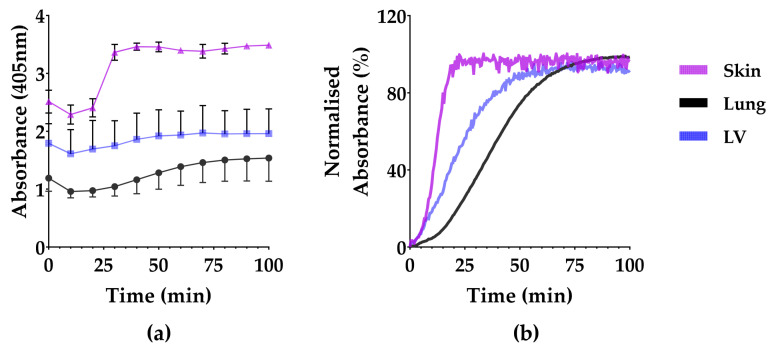
Turbidity of skin-ECM, lung-ECM and LV-ECM hydrogels. (**a**) Turbidity data (mean and S.D.) at every 10 min; (**b**) Normalized absorbance from the start of gelation. All data derived from a minimum of three independent experiments performed in triplicates. Data are presented as mean with SD.

**Figure 3 polymers-13-03113-f003:**
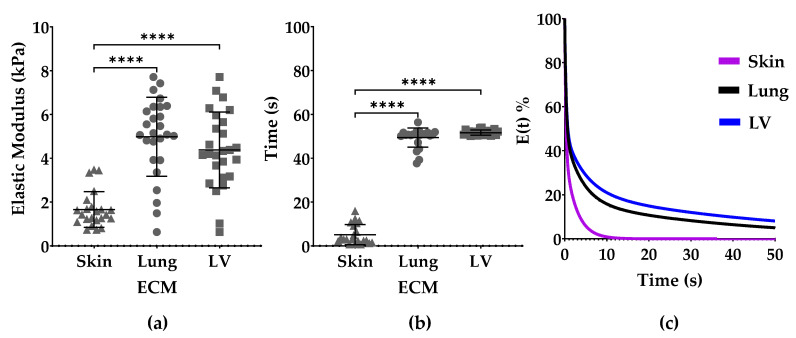
Viscoelastic properties of skin-ECM, lung-ECM and LV-ECM. (**a**) Elastic modulus; (**b**) Time to reach 50% stress relaxation; (**c**) Average stress relaxation normalized to start from 100%. All data derived from a minimum of three independent experiments performed in triplicates from low-load compression testing at 0.2 strain. Data are presented as mean with SD. Statistical differences according to one-way ANOVA and Dunn’s post hoc test **** *p* < 0.0001.

**Figure 4 polymers-13-03113-f004:**
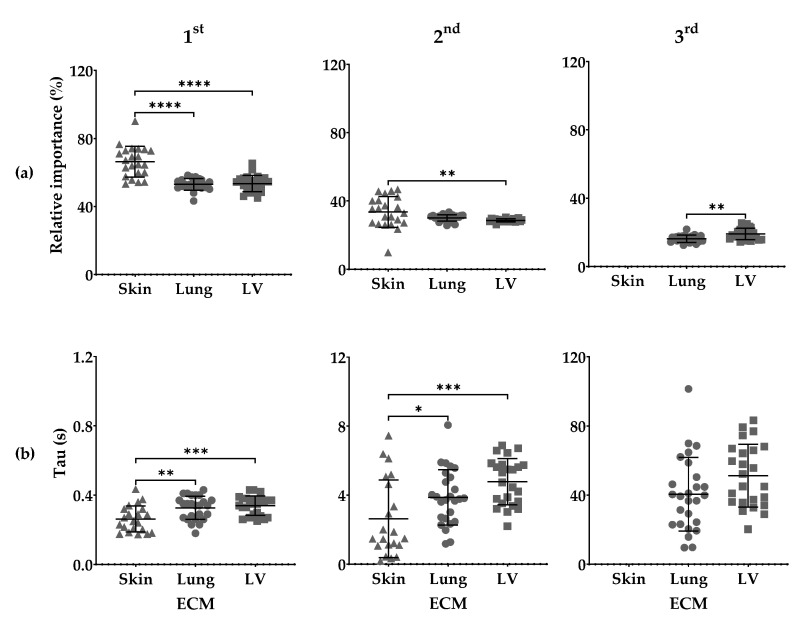
Maxwell analysis of skin-ECM, lung-ECM and LV-ECM viscoelasticity. (**a**) Relative importance of 1st, 2nd and 3rd Maxwell elements; (**b**) Tau (*τ*) of 1st, 2nd and 3rd Maxwell elements reported in seconds (s). All data derived from a minimum of three independent experiments performed in triplicates. Data are presented as mean with SD. Statistical differences according to one-way ANOVA and Tukey (1st and 2nd Maxwell Elements) and Student’s *t*-test (3rd Maxwell Element) * *p* < 0.05; ** *p* < 0.01; *** *p* < 0.001 and **** *p* < 0.0001.

**Figure 5 polymers-13-03113-f005:**
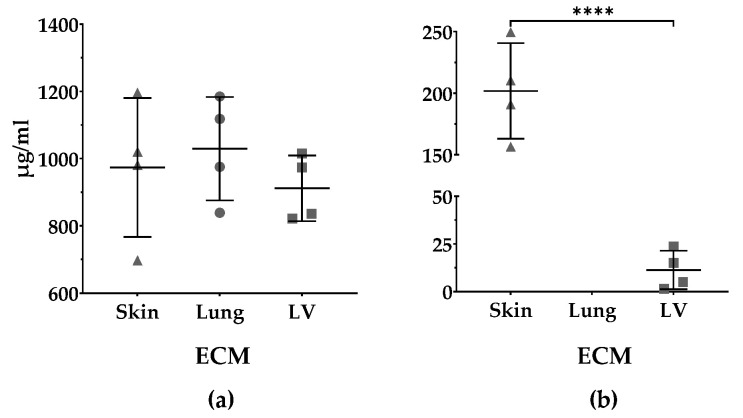
Quantification of protein and sulfated GAGs (sGAGs) of organ-derived ECM. (**a**) Protein content (µg/mL) of skin-ECM, lung-ECM and LV-ECM according to Lowry assay; (**b**) sGAGs content (µg/mL) skin-ECM and LV-ECM and according to DMMB assay. No sGAGs were detected in lung-ECM. All data derived from four independent experiments performed in triplicates. Data are presented as mean with SD. Mean values per experiment (n = 4) are also shown. Lowry data analyzed with one-way ANOVA (*p* = ns). DMMB data analyzed with Student’s *t*-test **** *p* < 0.0001.

**Figure 6 polymers-13-03113-f006:**
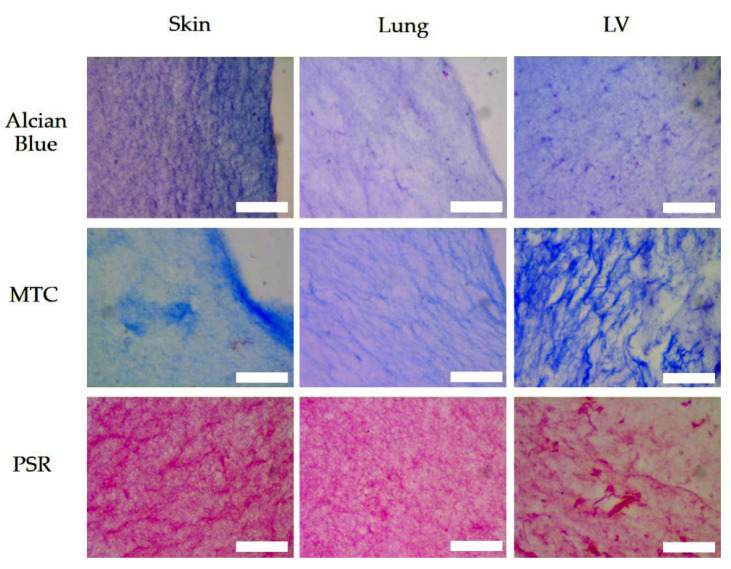
Alcian Blue, MTC and PSR, stains in sections of skin-ECM, lung-ECM and LV-ECM. Scale bars: 50 µm.

**Figure 7 polymers-13-03113-f007:**
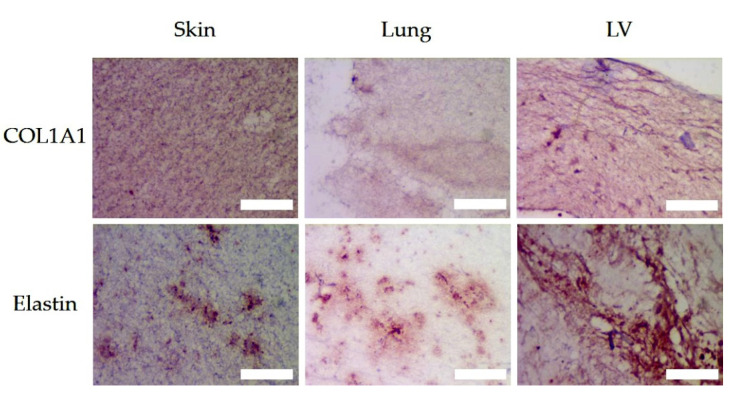
Immunohistochemistry (red) of COL1A1 and Elastin of skin, lung and LV ECM hydrogels. Scale bars represent 50 µm.

**Figure 8 polymers-13-03113-f008:**
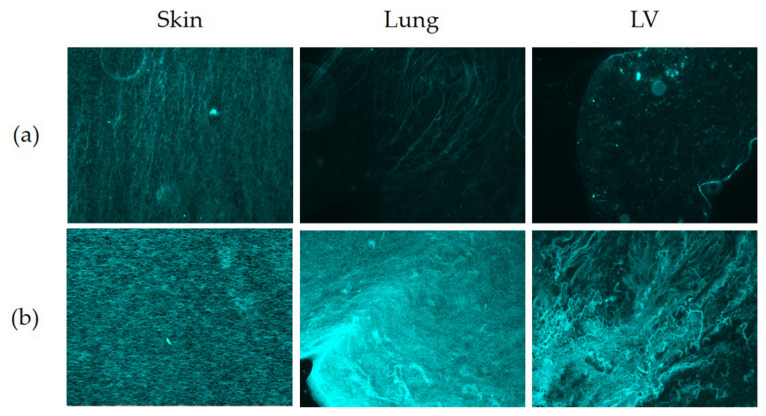
Fluorescent micrographs of PSR stained skin, lung and LV ECM hydrogels. (**a**) Unstained sections (autofluorescence); (**b**) PSR-stained sections. Original objective magnification 40×.

**Figure 9 polymers-13-03113-f009:**
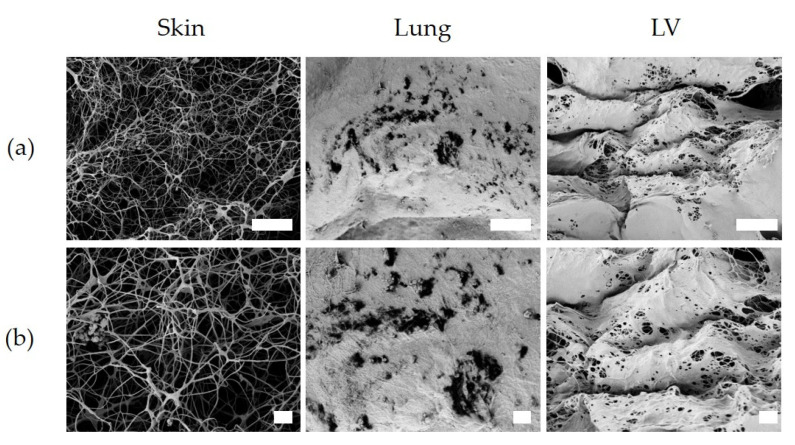
Surface microarchitecture of skin-ECM, lung-ECM and LV-ECM. (**a**) Magnification at 5000×; (**b**) Magnification, 10,000×. Scale bars: 10 (**a**) and 2 µm (**b**).

**Table 1 polymers-13-03113-t001:** Summary of Gelation kinetics parameters for organ derived ECM hydrogels ^1^.

Organ ECM	*A_min_*	*A_max_*	*T_lag_* (min)	*T*_1/2_ (min)	*T_end_* (min)	S (%/ min)
Skin	2.3 ± 0.2 ****^b^	3.5 ± 0.0 ****^b^	13.1 ± 2.9	12.2 ± 2.9 ****^b^	16.9 ± 3.2 ****^b^	5.3 ± 0.8 ****^b^
Lung	1.0 ± 0.1 ****^c^	1.4 ± 0.2 ****^c^	10.4 ± 4.7	43.2 ± 16.8 ****^c^	94.1 ± 11.7 *^c^	1.2 ± 0.3
LV	1.5 ± 0.1 ****^a^	1.8 ± 0.2 ****^a^	10.9 ± 6.5	20.9 ± 4.3 **^a^	75.7 ± 17.5 ****^a^	1.5 ± 0.6 ****^a^

^1^ All data is shown as Mean ± standard deviation (S.D.). Statistical differences between (a) skin-ECM, (b) lung-ECM and (c) left ventricle (LV)-ECM are highlighted and their significance shown: * *p* < 0.05, ** *p* < 0.01 and **** *p* < 0.0001; according to one-way ANOVA and Tukey after robust regression and outlier removal (ROUT).

**Table 2 polymers-13-03113-t002:** Results from TWOMBLI Analysis.

COL1A1								
Organ ECM	Number of Fibers	^1^ Mean Fiber Length	^1^ Total Fiber Length	Fiber Alignment	Number of Branch Points	Number of End Points	Lacunarity	^2^ HDM^1000%
Skin	15.3 ± 1.6	25.3 ± 0.5	8504 ± 2064	0.19 ± 0.04	121.3 ± 81.8	552 ± 95	14.7 ± 0.8	82.7 ± 44.0
Lung	14.9 ± 0.7	25.4 ± 1.2	28,316 ± 17,989	0.11 ± 0.07	350.7 ± 254.0	1902 ± 1179	16.7 ± 3.1	0.9 ± 0.1 *^a^
LV	13.9 ± 1.6	24.3 ± 1.2	15,337 ± 785	0.11 ± 0.01	158.7 ± 39.0	1106 ± 45	21.6 ± 4.7	4.2 ± 2.7 *^a^
**Elastin**								
Skin	10.7 ± 1.7 *^c^	19.3 ± 2.3 *^c^	2852 ± 270 **^c^	0.04 ± 0.01	27.0 ± 7.8 *^c^	273 ± 55c ***^c^	95.2 ± 15.6 ***^c^	9.1 ± 6.1 **^c^
Lung	11.7 ± 0.7	26.0 ± 2.7	9349 ± 1330 *^c^	0.04 ± 0.04	95.7 ± 23.2 *^c^	808 ± 153 **^c^	98.5 ± 16.0 ***^c^	11.2 ± 0.9 **^c^
LV	16.09 ± 2.60	26.0 ± 2.7 *^a^	17,869 ± 5099	0.09 ± 0.12	259.0 ± 109.7	1101 ± 169	18.5 ± 1.7	89.5 ± 33.0
**Picrosirius Red (Fluorescent)**								
Skin	14.19 ± 1.08 *^b^	27.8 ± 1.9 *^b^	33,481 ± 20,453 *^b^	0.07 ± 0.01	54.0 ± 41.7	2322 ± 1382 **^b^	95.4 ± 32.4 ****^b^	933.5 ± 35.1
Lung	16.71 ± 2.61	32.5 ± 5.0	18,818 ± 2972	0.07 ± 0.04	35.0 ± 16.7	18,818 ± 2972 *^c^	530.0 ± 129.0	31.1 ± 7.0 ****^a^
LV	12.66 ± 1.11 **^b^	24.7 ± 2.1 **^b^	30,346 ± 11,839 **^b^	0.07 ± 0.02	62.5 ± 26.8	30,346 ± 11,839 ****^a^	136.4 ± 29.0 ****^b^	59.3 ± 35.7 ****^a^

^1^ Data in micrometers (µm). ^2^ High Density Matrix (%). All data are shown as Mean ± standard deviation (S.D.). Statistical differences between (a) skin-ECM, (b) lung-ECM and (c) LV-ECM are highlighted and their significance shown: * *p* < 0.05; ** *p* < 0.01; *** *p* < 0.001 and **** *p* < 0.0001 according to one-way ANOVA and post hoc Tukey after robust regression and outlier removal (ROUT).

## Data Availability

Data are contained within the article and are available upon reasonable request.

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
