# Peer review of "Architecture and Composition Dictate Viscoelastic Properties of Organ-Derived Extracellular Matrix Hydrogels"

_polymers, 2021, doi:10.3390/polym13183113_

Round 1

Reviewer 1 Report

Dear authors, Dear Editor,

In the manuscript entitled “Architecture and composition dictate viscoelastic properties of organ-derived extracellular matrix hydrogels” Harmsen and co-workers study the vicoselastic properties of hydrogels derived from decellularised skin, lung and cardiac tissues. The authors establish a correlation between  the physical data (number of Maxwell elements and the relaxation time constant for each individual element time duration) and the protein (collagen and elastin) and sulfated glucosaminoglycan composition and  content and the micro-nano fibrilar architecture of the hydrogels.

This study suggests that ECM hydrogels derived from decellularised skin, lung and cardiac tissue (and probably other tissues) are suitable model systems for studying the dynamic  interplay between cell-ECM interactions and ECM composition, biomechanical  properties and cell fate.

The manuscript is very well written, clear and objective. The experimental work is performed to high technical and scientific standards and well-presented and discussed. The experimental methods are described with enough detail to permit reproduction by interested researchers.

This work is likely to appeal to the broad community of (nano)biomedicine researchers, especially those interested in tissue engineering and related areas.

The manuscript is clearly worth publication.

Minor points:

  • Please explain how was performed the normalization of the data of Figure 2a. Was the lag time omitted in the normalization?
  • The raw data relative to Figure 3 (stress-strain curves) and Figure 4 (E(t) curves) must be presented as “supplementary material”.

Author Response

Reviewer 1

Specific comments:

  1. Please explain how was performed the normalization of the data of Figure 2a. Was the lag time omitted in the normalization?

Answer: We agree with the reviewer that this explanation was missing in the figure legend. Indeed, the lag time was omitted for normalizing when generating Figure 2a. We explained this in line 158-159 with the description “The Normalized curved were plotted to start from gelation”. We added “omitting the lag time” description of this normalization in the materials and methods (line 159). In the figure legend we added “from the start of gelation” to line 291.

The raw data relative to Figure 3 (stress-strain curves) and Figure 4 (E(t) curves) must be presented as “supplementary material”.

Answer: As per the reviewer’s request, we have added the raw data for figure 3 and figure 4 to the supplementary materials. Supplementary table 1 contains all the values for the Elastic modulus, Time to reach 50% stress relaxation, Maxwell element Relative importance and Maxwell element Tau. The raw Tau and relative importance data in that table can be used to generate the stress relaxation curves in figure 3C with the use of equation 2 described in the materials and methods.

Reviewer 2 Report

This study investigated the viscoelastic properties of ECM hydrogels derived from organs and mathematically modelled these data with a generalized Maxwell model. The hydrogels gels provide opportunities for simulating the organ or tissue microenvironment, enabling the generation of novel models for mimicking native organ ECM in a research environment. Therefore, I recommend the publication of this manuscript after the following minor revisions.

  1. At line 6 in the abstract, ‘frm’ -> ‘from’.
  2. Water contents of the hydrogels should be evaluated.
  3. Some future remarks of this study should be added in the conclusion.
  4. English throughout the text should be checked.

Author Response

Reviewer 2

Specific comments:

  1. At line 6 in the abstract, ‘frm’ -> ‘from’.

Answer: We thank the reviewer for indicating this discrepancy, which we have changed in line 6 in the revised version of the manuscript.

  1. Water contents of the hydrogels should be evaluated.

Answer: The water content between the different organ derived ECM hydrogels are the same at the start of the experiment. The starting protein content is the same (20/mg) before digestion and afterwards as the protein assay shows we still have a similar protein content. In the making of the hydrogels the same volumes of NaOH and PBS are added.

However, we agree with the reviewer that the swelling of the different ECM hydrogels might be different. We would have liked to add this but we would not have enough time to add this to properly. We have added this as a limitation to our study (Line 495-501) in the discussion.  

  1. Some future remarks of this study should be added in the conclusion.

Answer: As per the request of the reviewer, we added some future remarks to the conclusion of the manuscript. Line 514 till line 519.

Reviewer 3 Report

This is quite a clearly-described study, demonstrating architecture and composition dictate viscoelastic properties of organ-derived extracellular matrix hydrogels. Overall, however, I judge this an interesting study which should be published in Polymers, subject to some minor revision as below.

  1. Line 24, what is (s)GAGs?
  2. Formats issue. Line 24, ‘(SEM’. Please check all.
  3. What are the main compositions of decellularized ECM from different tissues? What are the molecular weight and polydispersity of these polymers?
  4. Proper table 1 caption should be added.
  5. Figure 8, it is better to add some color to the fluorescent images.

Author Response

The answer to the reviewer 3 is in the documents
